# Ultrasonographic Renal Subcapsular Thickening in Cats with Primary and Metastatic Carcinoma

**DOI:** 10.3390/vetsci11030134

**Published:** 2024-03-20

**Authors:** Ayano Masuyama, Atsushi Toshima, Asami Nakajima, Masahiro Murakami

**Affiliations:** 1Department of Veterinary Clinical Sciences, College of Veterinary Medicine, Purdue University, West Lafayette, IN 47906, USA; 2Japan Small Animal Medical Center, Tokorozawa 359-0023, Japan; 3Nakajima Animal Hospital, Ayase 252-1104, Japan

**Keywords:** subcapsular thickening, kidney, carcinoma, metastasis

## Abstract

**Simple Summary:**

Renal subcapsular thickening is an ultrasonographic finding often related to lymphoma or feline infectious peritonitis in cats. Although a previous study reported that other renal neoplasia may cause subcapsular thickening, detailed information is lacking. Therefore, the purpose of this study is to describe ultrasonographic findings in renal subcapsular thickening and renal parenchyma in cats diagnosed with primary and metastatic carcinoma in the kidney. Nine kidneys from six cats met the inclusion criteria, including one primary renal carcinoma, four metastatic carcinomas, and four presumed metastatic carcinomas in kidneys from primary pulmonary carcinomas. In our study population, metastatic subcapsular thickening lesions were relatively thin, focal, homogeneous, and hypoechoic, whereas the primary ones were circumferential and heterogeneous. Additionally, hyperechoic renal parenchyma was observed in seven kidneys, three of which had concurrent hypoechoic striations. This is the first report documenting ultrasonographic subcapsular thickening in feline kidneys affected by metastatic carcinoma. Renal metastases, especially those arising from pulmonary carcinoma, should be included in the differential diagnosis when subcapsular thickening with the aforementioned ultrasonographic features is observed. The ultrasonographic findings of primary renal carcinoma may vary and require further investigation.

**Abstract:**

Ultrasonographic subcapsular thickening caused by renal neoplasia other than lymphoma has been previously reported in cats; however, detailed information is lacking. This study aims to describe ultrasonographic findings in renal subcapsular thickening and renal parenchyma in cats diagnosed or presumed with primary and metastatic carcinoma in the kidney. Ultrasound reports were retrospectively searched from 3 veterinary hospitals and 6 cats with 9 affected kidneys were included. Renal lesions were confirmed either cytologically or histopathologically as primary renal carcinoma with metastasis in the contralateral kidney (in 1 case), or metastatic pulmonary carcinoma (in 3 cases). Two patients were cytologically diagnosed with pulmonary carcinoma with concurrent renal subcapsular thickening. Eight kidneys affected by metastatic carcinomas showed relatively thin, focal, and homogeneously hypoechoic subcapsular thickening while a single kidney affected by primary renal carcinoma showed markedly thick, circumferential, and heterogeneously mixed iso- to hypoechoic lesion. The renal parenchyma, especially when just beneath the subcapsular lesion, exhibited at least one abnormality in all affected kidneys, most characterized by hyperechoic renal cortex with concurrent hypoechoic striations. This is the first report describing metastatic carcinoma causing renal ultrasonographic subcapsular thickening in cats. Our results suggest that renal carcinoma should be included in differential diagnoses when ultrasonographic subcapsular thickening is present in cats.

## 1. Introduction

Renal hypoechoic subcapsular thickening, also known as hypoechoic subcapsular rim, is an ultrasonographic finding in cats. It is characterized by a hypoechoic, either circumferential or crescent-shaped band along the renal margin beneath the renal capsule [1]. This finding is most commonly associated with feline lymphoma [1] and feline infectious peritonitis (FIP) [2,3].

Renal neoplasia is relatively rare in domestic animals, accounting for less than 1.7% and 2.5% of all tumors in dogs and cats, respectively [4,5,6]. The most common neoplasia in the feline kidney is infiltrating neoplasia, such as lymphoma, and the primary renal neoplasia is less common [7,8,9]. Renal cell carcinoma is the most common primary renal neoplasm in cats, accounting for more than half of primary renal neoplasms in cats [8,9,10,11,12]. Other primary feline renal neoplasms are reported to occur far less frequently. Previous reports have described carcinoma, nephroblastoma, adenoma, sarcoma, leiomyoma, carcinosarcoma, and hemangiosarcoma in the feline kidney [7,8].

Feline renal carcinoma is predominantly reported in middle-aged to older cats, with no apparent sex or breed predisposition [5,7,13,14]. Typically, renal carcinoma manifests as a solitary nodule or mass unilaterally located in either the left or right kidney [4,5,9], although instances of multiple or bilateral renal carcinoma have also been documented [5,15,16]. The characteristic appearance of renal carcinoma is that of a spherical or ovoid mass situated at one pole of the kidney, with clear demarcation from the remaining renal tissue [4,5,8,14]. In a previous study, renal masses were identified in 93% of canine renal carcinoma cases [17] and in 72% of feline cases [9], either through abdominal ultrasound or computed tomography (CT). A few cases of primary renal neoplasia, including undifferentiated malignant neoplasia and renal anaplastic carcinoma, have been reported to cause hypoechoic subcapsular thickening of the kidney in cats [1]. This study described the subcapsular thickening as a crescent-shaped hypoechoic area, or as a circumferential, rim-like hypoechoic area [1], but further detailed description is lacking.

The kidney is known to be a common site for metastatic tumors due to its high blood flow and rich capillary network [4,8]. Disseminated neoplasms of any type are likely to localize the kidney [8]. However, there is limited information on renal metastatic neoplasia in cats. Feline lung tumors frequently metastasize to various organs, including skeletal muscle, skin, liver, spleen, brain, intestine, bone, eye, and kidney [10,11,12,16,18]. Although intrathoracic metastasis is more likely to happen, the kidney is a relatively common extrathoracic metastatic site in cats with pulmonary carcinoma, with a prevalence of 1.5–15.3% of all metastatic cases [10,11,12]. Carcinoma is the primary cause of nodular renal metastasis in domestic animals, with pulmonary carcinoma occupying the majority [19]. Although nodular form of renal metastasis has been reported in cats with pulmonary carcinoma [11,19], ultrasonographic subcapsular thickening has not been described in previous reports as one of the manifestations of metastatic carcinoma in the feline kidney.

Therefore, the purposes of this study were (1) to describe the ultrasonographic features of primary and metastatic carcinoma in the feline kidney associated with ultrasonographic renal hypoechoic subcapsular thickening, and (2) to report the ultrasonographic changes observed in the renal parenchyma of these cats.

## 2. Materials and Methods

### 2.1. The Study Design and Patient Inclusion

This was a retrospective, multi-institutional, observational, descriptive study. Ultrasound reports from one veterinary university referral hospital in the United States and two primary and referral veterinary hospitals in Japan were retrospectively searched for cats with ultrasonographic subcapsular thickening in either one or both kidneys over a five-year period from 2019 to 2023. The use of medical record data was approved by the hospital owner/director and animal owners. Informed consent for the use of medical records for potential future retrospective study was obtained from owners through written consent forms and verbal consent at the time of presentation. Cats with cytologically or histologically confirmed diagnoses of primary or metastatic carcinoma in the kidney, or histologically diagnosed carcinoma in organs other than the kidney with ultrasonographic renal subcapsular thickening were included. Cats with poor-quality renal ultrasound images were excluded from the study. Inclusion and exclusion decisions were made by an ACVR board-certified radiologist (M.M.).

### 2.2. Medical Records Extraction

Medical records of the cats were reviewed and data on age, sex, breed, body weight, clinical complaints, and cytological or histological diagnoses were recorded. Complete blood count (CBC), and serum biochemistry results were also reviewed if available.

### 2.3. Ultrasonographic Evaluation of Subcapsular Thickening

Renal ultrasound images were obtained using linear array transducers (5–18 MHz). Depending on facilities, either ultrasound machine, ARIETTA 70 (Hitachi ALOKA Medical Corporation, Mitaka, Japan), ARIETTA 65 (FUJIFILM Healthcare Corporation, Tokyo, Japan), or Aplio i800 (Canon Medical Systems, Tustin, CA, USA) was used. An ACVR board-certified radiologist (M.M.) reviewed the renal ultrasound images and assessed subcapsular thickening and renal parenchyma. Ultrasonographic changes of subcapsular thickening were evaluated based on the following criteria: maximum thickness (measured in mm), distribution (focal, multifocal, or circumferential), echotexture (homogeneous or heterogeneous), echogenicity relative to the adjacent renal cortex (hyperechoic, isoechoic, or hypoechoic), renal cortical margin (smooth or irregular), capsular margin (smooth or irregular), vascularity (if color Doppler images were available), and presence of perirenal effusion (yes or no). Maximum thickness was defined as the longest distance perpendicular to the renal cortical margin, from the outer margin of the cortex to the renal capsular margin. The interface between the thickened subcapsular layer and the renal cortex was considered as the renal cortical margin, and the outer margin of the renal capsule as the renal capsular margin.

### 2.4. Ultrasonographic Evaluation of the Rest of the Kidney

Additional ultrasonographic features of the rest of the kidney were also assessed. The renal parenchyma was evaluated based on the following criteria: echotexture (normal or abnormal), corticomedullary distinction (well-defined, ill-defined, or completely effaced), renal pelvis diameter (measured in mm), presence of mineralization (yes or no), presence of cystic changes (yes or no), and presence of nodule(s) (yes or no). The echotexture of the renal cortex was subjectively evaluated, and it was considered abnormal if the normal renal parenchymal architecture was deformed or ill-defined. If any distinctive patterns of abnormal renal parenchyma were observed, they were described. Additionally, the renal pelvis diameter (mm) was measured using the transverse plane of the kidneys at the level of the renal pelvis.

### 2.5. Ultrasonographic Measurements of the Kidneys

In the present study, kidney size was defined as the length of the maximum longitudinal axis of the kidneys and measured using the sagittal plane of the kidney. The subcapsular thickening layer was included as a part of the kidney if it was present on the longitudinal axis. Kidneys were evaluated as small if the maximum longitudinal axis was less than 3.0 cm, and renomegaly was determined if the maximum longitudinal axis exceeded 4.5 cm [20].

## 3. Results

### 3.1. Study Population

A total of six cats met the inclusion criteria. None of the cases was excluded from the study. The signalment and diagnoses of the included cats are listed in Table 1. The study population consisted of three castrated male and three spayed female cats, with an age range of 9 to 15 years (mean ± SD, 11.7 ± 2.1 years). The breed of all included cats was Domestic Shorthair. The median body weight of the cats was 4.2 kg (range, 3.0–6.1 kg).

### 3.2. Clinical Data

The clinical complaints of cats with subcapsular thickening include hyporexia (5/6), lethargy (3/6), and weight loss (2/6). Multiple clinical complaints were recorded for each cat, resulting in a total number of complaints greater than the number of cats in the study. 

Histological or cytological diagnoses of renal subcapsular lesions were as follows: In one cat (case 6) undergone necropsy, primary renal carcinoma was diagnosed in one kidney, with metastasis observed in the contralateral kidney; both kidney lesions were confirmed by histopathology. Histopathologic examination revealed fibrosis, necrosis, and neoplastic cell infiltration involving the right kidney and expanding the renal subcapsular space. These neoplastic cells were polygonal with abundant eosinophilic cytoplasm, occasionally containing large clear vacuoles peripheralizing the nucleus, interpreted as Melamed–Wolinska bodies. These findings established the diagnosis of renal carcinoma for the right kidney. In contrast, the left kidney showed no gross abnormalities and no renal nodules or masses on examination. However, neoplastic cells indicative of metastasis were identified in the renal pelvis. This finding led to the diagnosis of metastatic renal carcinoma in the left kidney stemming from the primary carcinoma in the right kidney. In three cats, pulmonary carcinoma with subcapsular renal metastases was diagnosed, with both lung and renal lesions confirmed by cytology or histopathology. Cytological samples were collected from the thickened subcapsular lesions in two cats (case 1, 4). These samples revealed the presence of malignant epithelial cells with characteristics identical to those obtained from the lung lesions. Since the necropsy was declined by the owner, a postmortem biopsy was performed, and samples were collected from the affected kidney in one cat (case 5). The histopathological assessment uncovered that the renal samples were primarily composed of necrotic tissue with a few aggregations of atypical epithelial cells. In addition, two cats were diagnosed with pulmonary carcinoma confirmed by cytology, and renal subcapsular thickening was presumptive for metastatic carcinoma.

CBC and serum biochemistry results were available for five cats. Four cats had unremarkable blood test results. One cat (case 5), who was diagnosed with pulmonary carcinoma and renal subcapsular metastasis, both histologically confirmed, had mild anemia (22.6%), marked leukocytosis (53,276/µL), azotemia (96.0 mg/dL), and an elevated serum creatinine level (2.9 mg/dL).

### 3.3. Ultrasonographic Findings of Subcapsular Thickening

In our study population, renal subcapsular thickening was observed bilaterally in three cats, unilaterally in the left kidney in two cats, and in the right kidney in one cat, for a total of nine kidneys from six cats. For the image evaluation, 1–15 images and 0–2 videos were available per kidney. The ultrasonographic findings and diagnoses are listed in Table 1. The ultrasonographic evaluation showed that the maximum thickness of the subcapsular thickening ranged from 1.0 to 13.2 mm (median 3.0 mm), with one kidney with primary renal carcinoma showing a markedly thickened lesion (13.2 mm), while the others (metastatic carcinoma) were less than 5 mm in thickness. The distribution of the subcapsular thickening lesion was focal in eight kidneys (8/9; Figure 1, all metastatic carcinomas) and circumferential in a single kidney (1/9; Figure 2, primary renal carcinoma). In all eight metastatic carcinomas, the kidney showed focal, homogeneous, and hypoechoic subcapsular thickening (Figure 1). In a primary renal carcinoma kidney with circumferential thickening, the subcapsular thickening was heterogeneously mixed isoechoic to hypoechoic relative to the renal cortex (Figure 2).

The renal cortical margin was smooth in five kidneys (5/9; Figure 3) and irregular in four kidneys (4/9; Figure 1, Figure 2 and Figure 4). The capsular margin was smooth in five kidneys (5/9; Figure 3) and irregular in four kidneys (4/9; Figure 1, Figure 2 and Figure 4). Blood flow was assessed in five kidneys using color Doppler. The vascularity was observed within the thickened subcapsular lesion in two kidneys (2/5). Perirenal effusion was only observed in one cat, bilaterally.

### 3.4. Ultrasonographic Findings of the Rest of the Kidney

Ultrasonographic findings of renal parenchyma are listed below. The renal cortex was evaluated as normal in two kidneys (2/9) and abnormal in seven kidneys (7/9). In those 7 kidneys with abnormal renal cortices, the renal cortices were diffusely hyperechoic in three kidneys (heterogeneous in two and homogenous in one, Figure 2), and focally hyperechoic in four (single in two and multifocal in two, Figure 4).

A distinctive pattern was observed in three out of seven abnormal renal cortices, which showed hypoechoic striations within the hyperechoic areas (Figure 4). These hypoechoic striations of the renal cortices appeared to be associated with subcapsular thickening, as most of them were located just underneath the subcapsular thickening areas. Corticomedullary distinction was evaluated as well-defined in one kidney (1/9), ill-defined in seven kidneys (7/9), and completely effaced in one kidney (1/9). The renal pelvis was very mildly dilated in two kidneys, of which the diameter was 1.3 and 1.9 mm, and the others showed no dilation. Small multifocal mineralization was observed in the renal cortices and along the renal diverticula in two kidneys, both of which belonged to the same cat. A renal cortical cyst was present in one kidney. Two kidneys from two cats, both of which were diagnosed as pulmonary carcinoma with renal metastasis, exhibited a renal nodule. These nodules were hyperechoic to the adjacent renal cortex, and their sizes were 5.6 mm and 9.3 mm.

### 3.5. Kidney Size

The length of the longitudinal axis of the affected kidneys was measured in all the affected kidneys. The kidney size ranged from 3.4 to 6.0 mm (median 4.1 mm) with four kidneys showing renomegaly, all of which were diagnosed with metastatic carcinoma. None of the kidneys were determined as small.

## 4. Discussion

The present study describes ultrasonographic subcapsular thickening in nine kidneys from six cats affected by primary or metastatic carcinomas in the kidney. Eight cats were diagnosed with pulmonary carcinoma and concurrent renal metastasis. On the other hand, only one cat showed respiratory signs, indicating that the clinical signs in these cats may not be respiratory. Cats with pulmonary neoplasia may present with respiratory signs such as coughing or tachypnea [12,18,21,22]. On the other hand, a previous study reported that the most common clinical sign was weight loss and that respiratory signs were seen in less than half of the affected cats [10,12]. Additionally, some studies concluded that 9–36% of cats did not show any clinical signs and pulmonary neoplasia was an incidental finding [21,22], consistent with our results. Therefore, the thorax should be examined in cats with ultrasonographic renal subcapsular thickening even in the absence of respiratory signs.

In the previous study, histopathological examination revealed the presence of neoplastic cells and necrotic tissue in the subcapsular thickening layer of the feline renal lymphoma [1]. In case 5 in the current study, a histopathological report described the presence of neoplastic cells and necrotic tissue in the subcapsular thickening area as well. The ultrasonographic hypoechoic subcapsular thickening is believed to be caused by the lymphatic drainage system of the kidney, as lymphatic capillaries are present between the capsule and the cortex [1]. The subcapsular lymphatics, which are rich in the lymphatic system, are located beneath the renal capsule [23]. Feline lung tumors often metastasize to various organs, including the kidneys, through blood and lymphatics [10,11,12,18]. Although in our study histopathological assessment of two kidneys with metastatic carcinoma (in cases 5 and 6) did not elucidate the specific route of metastasis, subcapsular space could be a site for metastasis in various malignant neoplasia. Further investigation is imperative to comprehend the underlying mechanism of subcapsular thickening formation.

Ultrasonographic renal subcapsular thickening is a distinctive finding in cats in veterinary medicine, primarily associated with conditions such as lymphoma or FIP [1,2,3]. In human medicine, a comparable phenomenon is described as a renal subcapsular hypoechoic rim or halo [24,25,26,27,28]. The causes of this finding in humans include renal artery occlusion [24], renal cortical necrosis [28], or renal lymphoma [25,26,27], corresponding to edematous perfused tissue fed by capsular collateral vessels, necrotic tissue, or infiltrating neoplastic tissue, respectively [29]. In our study, the histopathological evaluation of the kidney with metastatic pulmonary carcinoma revealed necrotic tissue and neoplastic cells in the subcapsular thickening layer of the kidney. Additionally, cytological samples from two kidneys affected by metastatic pulmonary carcinoma indicated the presence of neoplastic cells in the subcapsular thickening lesions. Conversely, in the kidney with metastatic renal carcinoma, the histopathological report made no mention of the subcapsular region. These findings imply that the ultrasonographic evidence of subcapsular thickening may result from either necrosis or neoplastic infiltration. However, it is important to note that a more thorough prospective comparison of ultrasonographic and histologic findings is needed to definitively determine the origin of hypoechoic subcapsular thickening.

Previous reports described the ultrasonographic subcapsular thickening as a hypoechoic, crescentic focal or circumferential diffuse lesion [1,2]. This finding was initially reported as a feature associated with feline lymphoma, with an 80.9% positive predictive value and a 66.7% negative predictive value [1]. Subsequent research showed that the subcapsular thickening was not exclusive to feline lymphoma, but could also occur in cats with FIP, with 20–33% of FIP cats exhibiting hypoechoic subcapsular thickening [2,3]. However, these reports did not include metastatic carcinoma as a cause of renal subcapsular thickening [1,2,3]. In the present study, eight kidneys from six cats showed renal subcapsular thickening with a confirmed or presumed diagnosis of metastatic carcinoma in the kidney. All the kidneys affected by metastatic carcinoma showed focal subcapsular thickening, which could be a characteristic of the subcapsular thickening caused by renal metastasis. However, due to the small number of cases in our study, further study is needed for verification.

In a previous study, renal subcapsular thickening in feline lymphoma was all hypoechoic and was reported to be a crescentic focal lesion in 81% and a circumferential diffuse lesion in 19% [1]. Although the localization of the lesion was not mentioned, hypoechoic subcapsular thickening was observed in approximately a quarter of the cats with FIP [2,3]. In the present study, we observed thin (less than 5 mm), focal, homogeneous, and hypoechoic renal subcapsular thickening in eight kidneys with confirmed or presumed metastatic carcinoma. Although previous studies lack detailed descriptions of the subcapsular thickening, the provided images in the previous studies appear to share some similarities with cats in the present study. Therefore, our results suggest that when subcapsular thickening with these ultrasonographic features is seen in cats, metastatic carcinoma should be considered as a potential differential diagnosis, as it may show similar ultrasonographic findings to those observed in cats with lymphoma and FIP.

Renal subcapsular changes similar to ultrasonographic renal subcapsular thickening have been documented in human patients using CT in conditions such as renal lymphosarcoma and Rosai–Dorfman disease [30,31,32]. Although CT findings have been described in a cat with renal lymphoma, the specific subcapsular changes were not detailed, although the cat had ultrasonographic hypoechoic renal subcapsular thickening [33]. To our knowledge, no previous studies have explicitly described renal subcapsular thickening in cats using imaging modalities other than ultrasonography.

The most common cause of renal metastases is reported to be pulmonary carcinoma, which accounts for 25% of renal metastases in domestic animals [19]. Although no study has specifically investigated the cause of renal metastasis in cats, several studies have reported that feline pulmonary carcinoma can metastasize to the kidneys, with a prevalence ranging from 1.2 to 15.3% [10,11,12]. In the present study, three cats were found to have cytologically or histologically confirmed metastatic carcinoma in the kidney. The remaining two cats with presumed metastatic carcinoma, had similar ultrasonographic findings with the confirmed diagnosis of pulmonary carcinoma, suggesting the same etiology. To the authors’ knowledge, this is the first published study demonstrating that feline pulmonary carcinoma can cause hypoechoic renal subcapsular thickening in cats.

Furthermore, a previous report on ultrasonographic renal hypoechoic subcapsular thickening included two cats with presumed primary renal neoplasia, specifically undifferentiated malignant neoplasia and renal anaplastic carcinoma [1]. These cats showed hypoechoic subcapsular thickening, but the report did not provide ultrasonographic images or a detailed description of these findings in cats with primary renal neoplasia [1]. In the present study, we observed subcapsular thickening in a single case of primary renal neoplasia in a cat, which showed heterogeneously mixed isoechoic to hypoechogenicity (Figure 2). Although this finding was based on only one case, our results suggest that the subcapsular thickening caused by primary renal neoplasia may not always show a diffuse hypoechoic appearance.

In the present study, ultrasound did not reveal the presence of nodule or mass formation in the kidney affected by primary renal carcinoma. Typically, renal carcinoma in animals is characterized by a spherical or ovoid mass located at one pole of the kidney [4,5,8,14]. Supporting this, a previous study found that 72% of cats with renal carcinoma exhibited renal masses on ultrasound [9]. Since this study did not perform surgery or necropsy for all cases, the actual percentage of renal carcinoma cases forming a mass remains unknown. Another study indicated that certain types of primary renal neoplasia, including renal anaplastic carcinoma, can induce subcapsular thickening in cats. However, this study did not clarify the presence or absence of renal nodules or masses [1]. In our investigation, necropsy and histopathological assessment confirmed the absence of renal nodules or masses in the kidney affected by renal carcinoma. This suggests that even without renal nodules or masses, renal carcinoma could be a potential differential diagnosis, particularly when subcapsular thickening is observed.

Most lymphoma cats with renal lymphoma were reported to have abnormalities including renomegaly, irregular shape, hyperechoic renal cortex, and pyelectasis [1,34,35]. Some FIP cats also showed renal abnormalities such as renomegaly, pyelectasis, irregular shape, abnormal echogenicity in the renal cortex and medulla, and decreased corticomedullary junction differentiation [2,3]. In the present study, kidneys showed abnormal renal cortical architecture including hyperechoic parenchyma, some with hypoechoic striation, which appeared to be associated with subcapsular thickening. As renal parenchymal striations caused by renal neoplasms have been documented [20], hypoechoic striations seen in the present study may be attributed to metastatic carcinoma, especially that of pulmonary carcinoma. In the present study, histological confirmation of two kidneys with metastatic carcinoma and one kidney with primary renal carcinoma was obtained from both the subcapsular thickening layer and renal parenchyma. Additionally, cytological confirmation of the other two kidneys with metastatic carcinoma was obtained through fine needle aspiration of the subcapsular thickening layer. However, due to the retrospective nature of the study and the small sample population, a detailed correlation between ultrasonographic renal parenchymal changes and histopathology was not possible. Further studies are necessary to investigate this aspect as well as the etiology of these findings.

Renal metastasis is relatively common in animals. Metastatic lesions in the kidney are often microscopic, involving bilateral renal cortices [8]. Previous studies revealed that kidneys with metastasis from pulmonary carcinoma exhibited multifocal nodules on the renal capsular surface or within the parenchyma [11,19]. In contrast, our study identified renal nodules in only two out of six kidneys with metastatic pulmonary carcinoma. Unfortunately, due to the lack of necropsy or histopathological evaluation of these kidneys in our cases, the actual prevalence of renal nodules remains unknown.

A previous study reported that 64% of primary feline renal neoplasms demonstrated metastases, with the lung being the most common site, followed by the liver, abdomen/peritoneum, adrenal gland, and ureter [7]. In our study, only one cat had primary renal carcinoma. Histopathological examination from necropsy revealed metastases from the primary renal carcinoma to the lung and contralateral kidney, which also showed subcapsular thickening.

This study has several limitations. First, the sample size is small, particularly with respect to cats with primary renal cancer. Primary renal neoplasia is rare in cats, accounting for less than 2.5% of all feline tumors [4,5]. Although the kidney is a known metastatic site for various neoplasms in cats, such as pulmonary carcinoma [19], renal metastasis remains relatively infrequent. Even pulmonary carcinoma, the most common cause of renal metastases, has a low metastatic rate ranging from 1.2 to 15.3% [10,11,12]. The limited number of cases in our study is likely due to rarity of this disease, and might not fully represent the findings of the subcapsular thickening caused by primary or metastatic carcinomas. Consequently, our findings are primarily descriptive, aimed at documenting these unique renal changes associated with primary or metastatic carcinoma, rather than establishing definitive characteristics. Second, concurrent degenerative renal changes may have influenced our assessment of renal parenchymal changes. Third, the true prevalence of ultrasonographic renal subcapsular thickening in feline carcinoma cases remains undetermined. Finally, the inclusion of two cats without cytologic or histologic confirmation of renal subcapsular lesions. This may raise the possibility of other concurrent diseases, such as lymphoma or FIP, which are considered less likely but cannot be completely excluded. Further research is needed to determine the prevalence and explore the spectrum of renal parenchymal and subcapsular changes in cats with primary or metastatic carcinoma.

## 5. Conclusions

In conclusion, ultrasonographic hypoechoic subcapsular thickening can be seen in feline kidneys caused by primary or metastatic carcinoma. Thin, focal, homogeneous, and hypoechoic renal subcapsular thickening may be caused by metastatic carcinoma, especially metastasis from pulmonary carcinoma. Thus, it may be important to consider carcinoma as a cause of renal subcapsular thickening, with additional thoracic imaging possibly needed to search for primary neoplasia.

## Figures and Tables

**Figure 1 vetsci-11-00134-f001:**
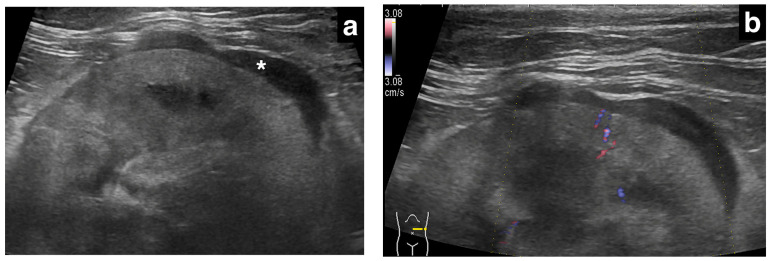
Ultrasonographic image of a kidney with metastatic pulmonary carcinoma (case 1, left kidney). A thin, homogeneously hypoechoic subcapsular thickening (*) was present focally along the ventral aspect of the kidney (**a**). There was no vascular flow observed within the subcapsular thickening using color Doppler (**b**).

**Figure 2 vetsci-11-00134-f002:**
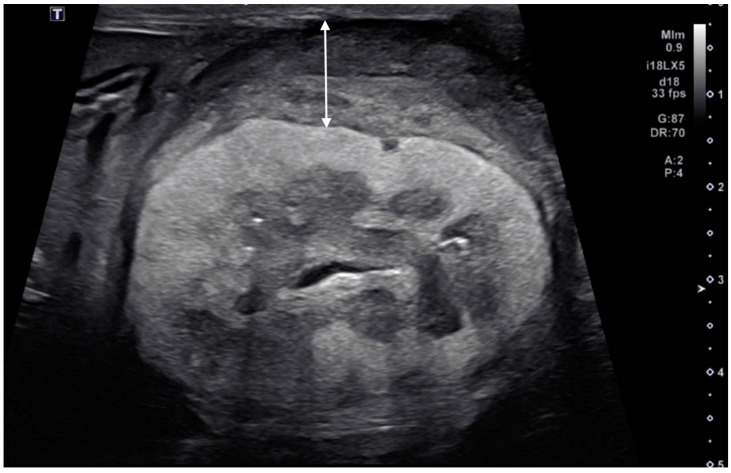
Ultrasonographic image of a kidney with primary renal carcinoma (case 6, left kidney). A thick, circumferential, heterogeneously mixed isoechoic to hypoechoic subcapsular thickening (double-headed arrow) was present. The renal cortical and the renal capsular margins were irregular. Note the diffuse, markedly hyperechogenic renal cortex.

**Figure 3 vetsci-11-00134-f003:**
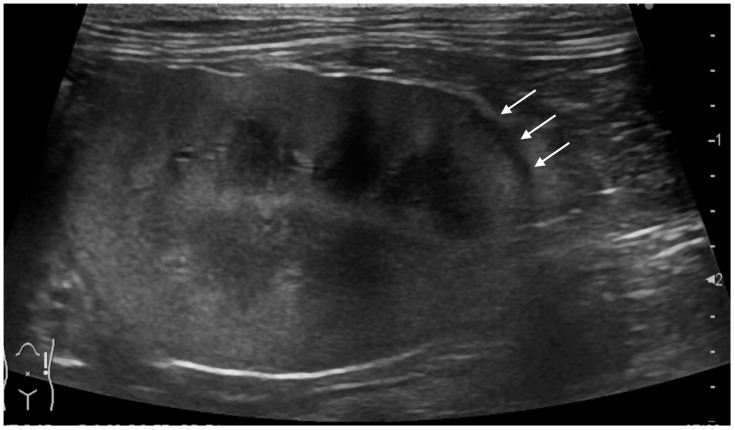
Ultrasonographic image of a kidney with metastatic pulmonary carcinoma (case 2, left kidney). A thin, homogeneously hypoechoic subcapsular thickening (arrow) was present. The renal cortical and renal capsular margins were smooth.

**Figure 4 vetsci-11-00134-f004:**
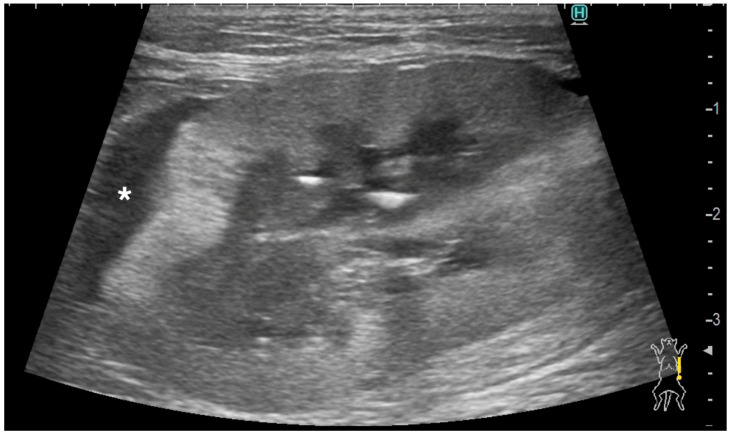
Ultrasonographic image of a kidney with metastatic pulmonary carcinoma (case 4, left kidney). A homogeneously hypoechoic subcapsular thickening (*) was present. The renal cortical and renal capsular margins were irregular. Hypoechoic striations were observed within the hyperechoic areas in the renal cortex.

**Table 1 vetsci-11-00134-t001:** The ultrasonographic findings of subcapsular thickening, renal parenchyma, and the rest of the kidney and diagnoses of nine kidneys from six cats.

Case	Signalment	Affected Kidney	US Findings of Subcapsular Thickening Lesion								US Findings of the Kidney							Diagnosis of Renal Lesion	Diagnosis of Other Organs
			Distribution	Maximum Thickness (mm)	Echotexture	Echogenicity	Renal Cortical Margin	Renal Capsular Margin	Vascularity	Perirenal Effusion	Architecture	Characteristic Pattern	Corticomedullary Distinction	Renal Pelvis Diameter (mm)	Mineralization	Cystic Lesion	Nodule		
1	10 year-old, MN, DSH	left	focal	3.5	homo	hypo	irregular	irregular	no	no	normal		ill-defined	1.3	no	no	yes	Metastatic Carcinoma and Suspected Metastatic Carcinoma (sample obtained from either right or left kidney)	Pulmonary Carcinoma (lung, muscle)
		right	focal	3.5	homo	hypo	irregular	irregular	no	no	abnormal	hypoechoic striations within hyperechoic areas	ill-defined	no dilation	no	no	no		
2	15 year-old, FN, DSH	left	focal	1	homo	hypo	smooth	smooth	NA	no	abnormal		ill-defined	no dilation	no	no	no	Suspected Metastatic Carcinoma	Pulmonary Carcinoma (lung, muscle)
3	11 year-old, MN, DSH	right	focal	2.8	homo	hypo	irregular	smooth	yes	no	abnormal		ill-defined	no dilation	no	no	yes	Suspected Metastatic Carcinoma	Pulmonary Carcinoma (lung)
4	9 year-old, FN, DSH	left	focal	4.8	homo	hypo	irregular	irregular	no	no	abnormal	hypoechoic striations within hyperechoic areas	ill-defined	no dilation	no	no	no	Metastatic Carcinoma	Pulmonary Carcinoma (lung)
		right	focal	1.5	homo	hypo	smooth	smooth	NA	no	abnormal		well-defined	no dilation	no	no	no	Suspected Metastatic Carcinoma	
5	14 year-old, FN, DSH	left	focal	1.2	homo	hypo	smooth	smooth	NA	no	abnormal	hypoechoic striations within hyperechoic areas	ill-defined	no dilation	no	no	no	Metastatic Carcinoma	Pulmonary Carcinoma (lung)
6	11 year-old, MN, DSH	left	focal	3	homo	hypo	smooth	smooth	NA	yes	normal		ill-defined	no dilation	yes	no	no	Metastatic Carcinoma	Metastatic Carcinoma (lung)
		right	circumferential	13.2	hetero	hetero	irregular	irregular	yes	yes	abnormal		completely effaced	1.9	yes	yes	no	Primary Renal Carcinoma	

Abbreviations: US, ultrasonographic; MN, male neutered; FN, female neutered; DSH, domestic short haired cat; homo, homogeneous; hetero, heterogeneous; NA, not applicable.

## Data Availability

The data that support the findings of this study are available from the corresponding author upon reasonable request.

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
