# Peer review of "Ultrasonographic Renal Subcapsular Thickening in Cats with Primary and Metastatic Carcinoma"

_vetsci, 2024, doi:10.3390/vetsci11030134_

Round 1

Reviewer 1 Report

Comments and Suggestions for Authors

Nice work and well presented. Congratulations to the authors.

I only have a question: do you have data from the renal flow in the kidneys? That could be an interesting point for evaluating the renal functionality and the chronicity of the illness. A Doppler analysis of the renal parenchyma, as you did in the subcapsular thickening, would be interesting to notify.

Author Response

Thank you for your encouraging comments and for the time you have invested in reviewing our manuscript. We greatly appreciate your insightful suggestion regarding the inclusion of renal blood flow data, specifically by Doppler analysis of the renal parenchyma.
Unfortunately, our retrospective data set did not contain renal parenchymal Doppler analysis. 

Given the importance of your suggestion, we are motivated to include renal flow data in our future studies. We believe that the inclusion of Doppler analysis in our upcoming projects will significantly contribute to our understanding of the disease processes under investigation.

Again, we thank you for your constructive feedback and look forward to incorporating this valuable aspect into our future research efforts.

Reviewer 2 Report

Comments and Suggestions for Authors

Dear Authors, 

I reviewed the manuscript entitled "Ultrasonographic Renal Subcapsular Thickening in Cats with Primary and Metastatic Carcinoma". The manuscript is well written and the topic very interesting, since subcapsular thickening in feline kidney is an uncommon peculiar finding, which deserves further investigation.

I have only some minor comments (see below), therefore I recommend minor revision.

Results

How many cases were excluded (if any)?

line 141: "In three cats, pulmonary carcinoma with subcapsular renal metastases was diagnosed, with both lung and renal lesions confirmed by cytology or histopathology." It would be interesting to show histological evaulation of subcapsular thickening in these cases.

line 146. Please, add more information on CBC and serum biochemistry. Did the cat with elevated serum creatinine and azotemia have severe renal changes? Was the only cat with serum biochemistry abnormalities?

Discussion 

lines 225-234: please discuss all limitations in a single paragraph. Move the paragraph to line 317.

line 278: please discuss histological findings of subcapsular thickening, if available. A dedicated figure could be useful to the reader too.

Reviewer 3 Report

Comments and Suggestions for Authors

Dear Authors, 

Thank you very much for your work. I really appreciated it. I had this week a very thick unilateral subcapsular thickening in a cat, I’m waiting for the FNA. Carcinoma was on my list of differential.

I also have a few comments to share with you. English is not my first language, so please forgive any mistakes in my writing style.

Introduction:

I would recommend rewriting the introduction in a different order to improve the reading. 

-Start with the most common neoplasia (Primary, infiltrating or metastatic) (now you 3rd paragraph)

-add a paragraph about the typical appearance of carcinoma (primary or metastatic) in cats on imaging

-continue with the subcapsular thickening mentioning all the neoplasia where it has been reported  (too superficially treated in the actual introduction line 56 and developed in the discussion L 239-341) and how it looks like on US. Finishing the paragraphs by the fact it has never been published to date in renal carcinoma (I or II) in cats

-and then finish with your objectives. I would remove the sentence line 69 to 71

Materials and methods

-line 82: you mentioned you had the animal owner approval. How did you get it retrospectively (phone, email, other?)

-part 2.3: can you add the ultrasound probes used and if the renal images were performed with microconvex or linear probe?

-part 2.4: how many images or videos were available for retrospective review per case? This is for me important data to provide to the reader.

-why the renal length was not assessed on ultrasound? Can you add it using the images available or the initial report?

-I would like you to remove the part on the radiographic measurement of the kidney. US should be used to measure the renal length, not ultrasound. This is irrelevant for your article to assess renal length with radiographs not in all patients, and it's not in your objectives

-I would like the US renal length to be included in the retrospective US evaluation of the kidney

-in part 2.4, considering you are looking at carcinoma, the criterion of cortical or medullary nodule is missing there. The absence of nodules or masses is also a very important finding of this article from my point of view.

Results

-part 3.3: need to tell how many images/videos were reviewed

-line 154: rather than giving the mean and SD, it will make more sense if you provide median

-Percentage: you have a so small group of cats, I would not use the percentages at all in your results

-Table 1: renal pelvis diameter is not a US finding of renal parenchyma. I would advise an extra column for the collective system. Another solution is to rename the column "US findings of renal parenchyma"

-Looking at the images you are providing, some of the subcapsular thickenings are deforming the cortex (images 2 and 4) whereas some are not (image 1). Could this observation (mass effect?) be included too

-line 200: can you describe more the striation there? You are doing it in the discussion where it's not its place.

-Line 205: replace by "A renal cortical cyst"

-remove lines 207 to 211 and replace it with the US renal length and thickness

Discussion

My general comments about this discussion are:

-This is too long

-There is too much repetition of the result and not enough discussion (for example lines 215 to 223)

-to my point of view, the discussion should follow the order of your results

-There are 2 paragraphs about limitations (Lines 225 to 233 and 317 to 327)

-I'm missing a comparison with other modalities in cats, and other species

Line per line comments:

-L243 to 249: this is a repetition of the results

-L252 to 255: I would add that for your metastatic cases, the capsular thickening was partial.

-line 259: why did you use "similarly"?

-L 261: if the subcapsular thickening reported in mm in cases or lymphoma, FIP or the other neoplasia reported?

-L267-268: renomegaly is not a renal parenchyma changes to my point of view

-L273-274: this is a result, not mentioned in the result (as previously mentioned)

-279: can you go back to the histology report available on the sample with subcapsular thickening and give more detail about that was the underlying tissue associated with the neoplastic cells? lymphatics? fibrous tissue? renal tissue?

-L 294 to 296: in these studies mentioned, did the authors mention the pathologic appearance of the metastasis (nodules/ masses/ infiltrate...) can you find in histopathological textbooks the normal appearance of renal metastasis in cats?

-L 301 to 305: I really like this part, but not sure it's position is right in the discussion

-L306 to 315: this is about the histology, but you briefly mentioned histology and cytology in a previous paragraph. Can you speak about histology/cytology at only one paragraph? This is too hypothetical, and needs to be correlated to the sample obtained on your cases.
